# All-Aerosol-Jet-Printed Carbon Nanotube Transistor with Cross-Linked Polymer Dielectrics

**DOI:** 10.3390/nano12244487

**Published:** 2022-12-19

**Authors:** Bhagyashree Mishra, Yihong Maggie Chen

**Affiliations:** 1Materials Science, Engineering, and Commercialization, Texas State University, San Marcos, TX 78666, USA; 2Ingram School of Engineering, Texas State University, San Marcos, TX 78666, USA

**Keywords:** thin-film transistor, polymer dielectrics, carbon nanotubes, completely printed flexible transistor, aerosol jet printing, flexible electronics

## Abstract

The printability of reliable gate dielectrics and their influence on the stability of the device are some of the primary concerns regarding the practical application of printed transistors. Major ongoing research is focusing on the structural properties of dielectric materials and deposition parameters to reduce interface charge traps and hysteresis caused by the dielectric–semiconductor interface and dielectric bulk. This research focuses on improving the dielectric properties of a printed polymer material, cross-linked polyvinyl phenol (crPVP), by optimizing the cross-linking parameters as well as the aerosol jet printability. These improvements were then applied to the fabrication of completely printed carbon nanotube (CNT)-based thin-film transistors (TFT) to reduce the gate threshold voltage (V_th_) and hysteresis in V_th_ during device operation. Finally, a fully aerosol-jet-printed CNT device was demonstrated using a 2:1 weight ratio of PVP with the cross-linker poly(melamine-co-formaldehyde) methylated (PMF) in crPVP as the dielectric material. This device shows significantly less hysteresis and can be operated at a gate threshold voltage as low as −4.8 V with an on/off ratio of more than 10^4^.

## 1. Introduction

Printed electronics have emerged as a low-cost alternative to silicon electronics and enable the manufacturing of electronic devices without high-temperature, -pressure, and -vacuum systems [1,2,3,4,5,6]. For flexible and large-area applications such as sensors, radio-frequency identification (RFID), and displays, printed electronics offer compatibility with various flexible substrates [7,8,9,10,11]. Moreover, most printed devices made of organic materials are biocompatible and have numerous applications in drug delivery, invasive sensors, skin electronics, wearables, etc. [12,13,14,15]. The main roadblock for printed electronics is their lower performance because of the resolution of the printing technique and the poor printability, availability, and compatibility of materials for solution processing or ink formation [16,17,18,19,20,21]. Yet, some materials, such as carbon nanotubes (CNTs), have shown extraordinary qualities in terms of electrical, mechanical, and thermal properties [22,23,24,25]. CNTs can be solution-processed for printed applications and are compatible with various other inks and flexible substrates. In recent years, tremendous progress has been made for printed CNT thin-film transistors (TFTs) to achieve a high on/off ratio of up to 10^6^ and mobility up to 25 cm^2^ V^−1^ s^−1^. However, the devices face challenges such as large hysteresis in gate threshold voltages, sub-threshold swings, and the availability of reliable printed dielectric materials [6,26,27]. Most previously printed CNT-TFTs have been fabricated by using atomic layer deposition (ALD) to deposit a thin layer of a dielectric (mainly metal-oxide-based) for high performance [28,29,30]. The hydroxyl group (-OH) present in these dielectrics acts as an interface trap at the junctions of CNTs and dielectric layers [31,32]. Amorphous dielectrics also suffer from bulk charge traps due to broken bonds that result in hysteresis during device operation [33,34]. These can be removed by high-temperature treatment under vacuum conditions. However, most oxide materials are incompatible with current printing techniques [29,35].

There are very few dielectric materials available that can be reliably printed using inkjet and aerosol jet printing. Most past research used ion gel as a printable dielectric material. Though the on/off ratio of the device is high, it is challenging to print ion gel, and it becomes unstable at high temperatures in ambient air, making the device unreliable [36,37]. Some research used a barium titanate/poly (methyl methacrylate) (BiTaO3/PMMA) nanoparticle–polymer matrix to print TFTs. For these types of dielectrics, the shape, size, and distribution of nanoparticles in the polymer greatly influence device performance, speed, and reliability [7,38]. Homenick et al. used BaTiO3/PMMA as a dielectric to print a CNT TFT using a combination of roll-to-roll gravure printing and inkjet printing [39]. They used polymer-wrapped CNT ink to avoid bundling and encapsulated the device to enhance its lifetime.

In recent years, the use of polymer dielectric materials has increased significantly. Most of them are PVA- or PVP-based materials. Most organic FETs use these polymer dielectrics and deposit them by spin coating. So, most of these TFTs are bottom gates, and clean-room techniques such as lithography are needed for further patterning. Cao et al. used one PVP-based material called xi-dcs (a blend of poly (vinyl phenol)/poly (methyl silsesquioxane) (PVP/pMSSQ)) to fabricate a fully aerosol-jet-printed hysteresis-free CNT TFT with a high on/off ratio [40]. However, the threshold voltage of the TFT was as high as −12.5 V because of the thick dielectric layer (~ 2 µm). Chen et al. optimized AJP parameters for the same dielectric material to fabricate a CNT TFT with a low threshold voltage within 10V [41]. One of the significant disadvantages of PVP- and PVA (polyvinyl alcohol)-based dielectric materials is the threshold voltage shift between subsequent measurements of transfer characteristics. As these polymers have free hydroxyl (-OH) groups, atmospheric water molecules are absorbed by the polar dielectric to form dipoles. Hence, these polymers must be cross-linked with another compatible polymer to reduce the number of free -OH bonds and enhance the insulating properties. The concentration of the cross-linker relative to the polymer and cross-linking parameters, such as temperature in thermal cross-linking, are critical parameters for device performance [42,43,44,45]. Poly(melamine-co-formaldehyde) methylated (PMF) has been the most successful when used as a cross-linking agent for PVP, and they are thermally cross-linked at different temperatures from 100 °C to 200 °C. Various studies have demonstrated TFTs with cross-linked PVP-PMF at different weight ratios, ranging from 1:1 to 2:1 to 5:1 [46,47,48,49]. Almost all experiments have used the spin-coating method to deposit the cross-linked PVP (crPVP) solution. Park et al. experimented with different weight ratios of PVP-PMF at different annealing temperatures to showcase hysteresis-free TFTs at a low temperature of 100 °C [50]. However, it is evident from other research that a higher annealing temperature offers the best cross-linking between PVP and PMF. We found that only Gao et al. used inkjet printing to deposit the cross-linked PVP-PMF solution at a 2:1 weight ratio for a CNT-based TFT [51]. They fabricated an electrolyte transistor by mixing an ionic liquid into a crPVP solution and enabled low-voltage operation. No studies to date have shown the aerosol jet printability of a crPVP solution and its optimization for low-voltage hysteresis-free CNT TFT operation.

In this work, a single technique, aerosol jet printing, was used to fabricate a CNT TFT using a cross-linked PVP-PMF dielectric material. Aerosol jet printing is compatible with a wide range of ink viscosity (1–1000) Cps and offers higher-resolution printing than inkjet printing. CNT networks were deposited utilizing layer-by-layer deposition to decrease bundling and rinsed with a solvent to remove the excess surfactant. crPVP solutions were prepared at different weight ratios of PVP to PMF and characterized for their use as the dielectric layer. The effects of various weight ratios of PVP to PMF in the crPVP solution on the hysteresis behavior of fully printed devices were compared and analyzed. A 2:1 wt. ratio of PVP to PMF was found to have less hysteresis in the gate threshold voltage for forward and reverse sweeping. Then, the AJP parameters were optimized to print a flexible pin-hole-free thin dielectric layer using the 2:1 wt. ratio crPVP solution for low-voltage operation (less than 10 V). This research will significantly contribute toward the use of polymer dielectrics for completely printed low-voltage operated transistors for low power dissipation. It will also add significant data on the best weight ratio of PVP to PMF to reduce hysteresis in printed transistors for reliable operation.

## 2. Materials and Methods

First, the dielectric ink was formulated by dissolving PVP and PMF in propylene glycol monomethyl ether acetate (PGMEA) and stirred overnight at room temperature. PVP and PMF were mixed at different weight ratios from 1:1 to 5:1 and different concentration gradients from 2.5 wt.% to 10 wt.%. Then, 10 wt.% solutions with various weight ratios of PVP to PMF were spin-coated at 2000 rpm for 60 s on a glass substrate, cured at 170 °C for 1 h, and checked by Fourier-transform infrared (FTIR) spectroscopy to measure the degree of cross-linking. The viscosities and contact angles of the solutions were measured to check the printability. After choosing the best ratios and concentrations, all solutions were tested for aerosol jet printability and then analyzed for pinholes in the dielectric film. For good dielectric films, capacitance–voltage (C-V) measurements were taken at different frequencies, and the dielectric constant was extracted at 100 KHz. Then, using the best dielectric film, flexible transistors were printed using the aerosol jet printer.

The proposed transistor consists of silver as the source, drain, and gate electrode, CNTs as the semiconductor, and crPVP as the gate dielectric. All of the devices were fabricated on Kapton using an Optomec AJ300 aerosol jet printer system (Optomec Inc., Albuquerque, NM, USA). Most of the materials received were used without any further modifications. Silver ink was obtained from Electroninks, Austin, TX, USA (EI-616), carbon nanotube inks were purchased from Raymore Nanotech, Quebec, Canada (IsosolS-100), and PVP, PMF, and PGMEA were received from Sigma Aldrich, St. Louis, MO, USA. The concentration of the received CNT ink was 20 mg/mL; for better jettability with the aerosol jet printer (AJ300), it was diluted to 0.01 mg/mL by adding toluene.

For device fabrication, first, the substrates were cleaned using IPA and acetone-bath sonication and then dried for 5 min using a hot plate. Before printing, all substrates were plasma-cleaned using 100 watts of oxygen plasma for 5 min. First, source and drain contacts were printed using silver ink, and the channel length was kept at 100 µm. The parameter settings for silver ink in AJ300 were 40 sccm for sheath gas, 15 sccm for atomization gas, and 600 mA current. Silver ink was cured at 140 °C in ambient air for 30 min. Then, CNT ink was printed on the channel between the source and drain contacts. To form a uniform CNT network, the substrates were exposed to oxygen plasma at 100 watts for 30 s. Then, CNT ink was printed in the channel between the source and drain with 50 sccm sheath gas, 12 sccm atomization, and 320 mA current. Before printing, the CNT ink was sonicated for 10 min to disperse all of the nanotubes uniformly in the solvent and prevent bundling. Three layers of CNTs were printed, with air drying of the ink and washing with toluene in between to remove excess surfactant. Previous research has shown that this layer-to-layer deposition of CNTs creates a uniform CNT network [42,43]. The CNT films were cured at 130 °C for 30 min. The crPVP dielectric ink was printed with 28–32 sccm sheath, 20–22 sccm atomization, and 600 mA current and cured at 100 °C for 1 min and 170 °C for 1 h for cross-linking. Finally, silver was printed as the gate electrode using the same process as the source and drain. The detailed fabrication process is shown in Figure 1. Figure 2a shows the printed device on Kapton, and Figure 2b shows the SEM image of the uniform CNT network after three layers of printing.

All of the devices were printed using an Optomec AJ300 aerosol jet printer. DC characterization of the transistor was performed using a four-point probe station and a Keysight semiconductor analyzer. Film thickness was measured using KLA Tencor P7 profilometer (KLA Tencor, Milpitas, CA, USA). The viscosity of the solutions was measured using a RheoSense m-VROC viscometer (RheoSense, San Ramon, CA, USA), and the contact angle was measured using a Kruss tensiometer (Kruss, Matthews, NC, USA). For contact angle measurements, dielectric materials with 10 wt.% concentration were spin-coated on the glass substrate and cured at 170 °C. FTIR spectroscopy was conducted using a Bruker ALPHA II FTIR Spectrometer (Bruker Co., Billerica, MA, USA), and C-V measurement of the dielectric film was performed using an 802–150 MDC Mercury Probe station (MDC, Chatsworth, CA, USA). All Scanning Electron Microscope (SEM) analyses were carried out using an FEI Helios Nanolab 400 system (FEI, Hillsboro, OR, USA). For SEM analysis, all materials were printed on a silicon substrate.

## 3. Results

### 3.1. PVP-PMF Weight Ratio Optimization (Effect of PVP-PMF Weight Ratio on Dielectric Properties and Printability)

Before printing the transistor, the formulation and printability of the crPVP ink were optimized. To investigate the optimal weight ratio of PVP to PMF, crPVP ink was formulated at different weight ratios of PVA to PMF (1:1, 2:1, 3:1, and 5:1). Previous research has shown that hydroxyl group species with various hydrogen-bonding interactions are responsible for the absorption of water molecules into the PVP dielectric, and that leads to increased leakage and hysteresis in transistors. The weight ratio and annealing temperature are essential factors in cross-linking to reduce the threshold voltage variation in TFT operation. It has been demonstrated previously that an annealing temperature of 170 °C or more forms a highly cross-linked film with lower hysteresis than in the other temperature range. So, all samples were annealed at 170 °C unless otherwise stated.

FTIR spectroscopy was conducted to measure the IR absorbance of the hydroxyl group present in PVA after cross-linking with PMF at various weight ratios. The wavenumbers associated with the hydrogen-bonded hydroxyl group, associated hydroxyl group, and non-hydrogen-bonded (free) hydroxyl group are ≈3340 cm^−1^, ≈3410 cm^−1^, and ≈3530 cm^−1^, respectively. From Figure 3a, it is evident that the intensity of the IR absorbance decreases with an increase in PMF content in the solution. The 1:1 weight ratio has the lowest absorbance peak for all of the wavelengths of different hydroxy groups and hence shows the highest cross-linking.

Then, to investigate the dielectric properties of PVP-PMF, metal–insulator–semiconductor (MIS) capacitors were fabricated using silver electrodes and the PVP-PMF dielectric with different weight ratios on a silicon wafer. The capacitance vs. voltage (C-V) curve of the dielectric with respect to their weight ratios at 100 KHz was measured. From the C-V curve, the dielectric constants were calculated depending on the thickness of the dielectric, and the graph is shown in Figure 3b. For this type of measurement, the instrumental uncertainty in the values of the dielectric constant is less than 1% for a single-layer specimen. It is observed that the 5:1 weight ratio of the PVP-PMF film has the highest dielectric constant for a 170 °C annealing temperature. For other films, the dielectric constant is slightly lower, with a variation of 18% from the 5:1 to 1:1 weight ratio. Molecular dipoles and mobile ions present in the dielectric bulk are responsible for the change in the dielectric constant, so a higher frequency of 100 KHz was chosen to minimize their effects and ensure that most charge carriers are electrons.

The printability of PVP-PMF is very crucial for fabricating fully printed TFTs. As the devices will be printed using the aerosol jet printing technique, various parameters, such as sheath gas flow (SG), atomization gas flow (UA), platen temperature, and printing speed (PS), need to be optimized to achieve a pinhole-free dielectric layer. Apart from printing parameters, ink formulation, curing conditions, substrates, the surface quality of the previously printed layer, and atmospheric moisture content also influence the quality of the dielectric film. In the AJ300 aerosol jet system, an ultrasonic atomizer is used to print the PVP-PMF ink, as it requires a much lower amount of ink and prevents ink wastage. The viscosity range for the ultrasonic atomizer is 1–5 cP. To achieve the viscosity range, first, the PVP-PMF inks were diluted to a 5 wt.%–2 wt.% concentration and checked for consistent jettability. For the initial experiments, the printing parameters used for all inks are a 20–24 sccm UA flow rate, a ~50 sccm SG flow rate, and 5–6 mm/s PS. For a low UA flow rate, it was hard to obtain a continuous ink stream; rather, the ink stream behaved like a spray of aerosols, as shown in Figure 4a. If the UA flow rate was too high, the volume of the ink increased and led to overspilling, as shown in Figure 4c. During printing, the UA flow rate was given more importance, while the SG flow was kept nearly constant because it is easier to control the thickness of the printed film by controlling the UA flow rate. For fixed UA and SG flow rates, the PS can be varied depending on the line thickness of the ink stream and the pitch of the design pattern. The PS should be under the machine acceleration limit for a certain pitch in the design pattern. Hence, 5–6 mm/sec PS was selected to provide the minimum needed overlap for continuity between subsequent printed lines and to obtain the minimum thickness at the same time for fixed UA and SG. As the 4 wt.% and 5 wt.% inks were more viscous, it was found that the ink needed a very high UA, above 30 sccm, to jet out at the highest atomization setting. The viscosity increases with an increase in PVP wt.; among the 5:1 PVP-PMF inks, the 4 wt.% and 5 wt.% inks did not form continuous lines, as shown in Figure 5b.

However, for the 2.5 wt.% concentration, PVP-PMF inks at all wt. ratios (2:1, 3:1, and 5:1) formed steady ink streams for 22 sccm UA and 50 sccm SG, as shown in Figure 5a for the 2:1 wt. ratio dielectric. Then, the PS was checked for the 2.5 wt.% ink to ensure overall coverage without voids or pinholes. One of the advantages of aerosol jet printing is that the UA, SG, and PS can be modified during printing to improve film quality. Finally, all PVP-PMF inks with various weight ratios (2:1, 3:1, and 5:1) at a concentration of 2.5 wt.% in PGMEA were printed at 22 UA, 50 SG, and 6 mm/s PS. These parameters were used to print the dielectric layer of fully printed CNT devices. In the initial experiment, to find the best weight ratios of PVP-PMF, all of the dielectric inks were printed with the same parameter settings for a fair comparison.

### 3.2. Performance of Printed TFTs Based on Different Weight Ratios

CNT-based TFTs were printed with various weight ratios of PVP to PMF (2:1, 3:1, and 5:1) as the dielectric layer using the same parameter settings and the same device structure. In this work, all of the devices were printed as top-gate TFTs to reduce hysteresis, and the CNT layer was printed on top of the silver source and drain to avoid the direct contact of carbon nanotubes with the environment and prevent the rapid degradation of the devices. The DC transfer characteristics, that is, the plot between Id (drain current) and Vgs (gate to source voltage), of devices with various weight ratios are shown in Figure 6 and Figure 7. From the transfer curves, it can be observed that it takes more positive gate voltage (Vth) to turn off the transistors with 3:1 and 5:1 weight ratios than the 2:1 wt. ratio TFTs. This is the result of a very thick dielectric layer, as 3:1 and 5:1 wt. ratio inks are more viscous and print thicker layers when printed using similar parameters. More diluted inks of 3:1 and 5:1 wt. ratios, such as those with concentrations less than 2.5 wt.%, will result in thinner dielectric layers. Additionally, the on/off ratios of TFTs printed using 3:1 and 5:1 have an on/off ratio of 10^3^, whereas most TFTs printed using a 2:1 weight ratio have an on/off ratio of 10^4^. This is because of the high off-current in 3:1 and 5:1 PVP-PMF dielectrics. For the TFTs with a 2:1 PVP-PMF dielectric, the threshold voltage is around 20 V for a 1–1.5-micron-thick dielectric.

However, we were more interested in hysteresis in the gate voltage during forward (on → off) and backward (off → on) sweeps. The hysteresis curves of TFTs in Figure 6 with various weight ratios clearly show that the 5:1 wt. ratio has the highest hysteresis in the gate threshold voltage, and it decreases with an increase in PMF content. So, the 2:1 PVP-PMF dielectric has the lowest hysteresis in Vth compared to that of 3:1 and 5:1, and this agrees with the conclusion of previous research.

In this case, there can be two main causes of hysteresis in the transistor performance: interface charge traps and bulk dielectric defects. Water molecules present in the atmosphere can be absorbed by the surfaces of dielectric materials having hydroxyl groups. Then, these molecules will diffuse into the bulk of the dielectric and create a polarization effect. Because of this, during the forward and reverse sweeps, we experience a change in the threshold voltage, which keeps increasing with the increase in the number of free hydroxyl bonds. Other researchers have also concluded that a higher PVP concentration in PVP-PMF mixtures leads to a higher free hydroxyl group concentration, which results in increased interface charge trapping between the semiconductor and dielectric layers. Figure 7 shows the variability analysis of the transfer characteristics of devices with different weight ratios. For each wt. ratio, three working devices were characterized for the Id-Vg curve with the same voltage and current settings. The variability in the gate threshold voltage and the on/off ratio is very low in all of the devices, irrespective of the weight ratio.

### 3.3. Demonstration of Optimized Device Characteristics

Finally, a 2.5 wt.% concentration of the PVP-PMF dielectric with a 2:1 weight ratio was chosen to fabricate a fully aerosol-jet-printed device on the flexible Kapton substrate with optimized printing parameters to lower the threshold voltage. The device threshold voltage was improved by controlling the thickness of the printed dielectric film. The greater the thickness of the dielectric layer, the higher the gate voltage required to operate the device. This is because of the dependence of the gate capacitance on the dielectric thickness, and the gate capacitance is related to transconductance. Figure 8 shows the positive correlation between the threshold voltage and dielectric thickness. Establishing a one-to-one correlation between the threshold voltage and dielectric thickness is difficult, as the threshold voltage also depends on the charge in the dielectric and operating temperature, and discussing these factors is outside the scope of this work. So, here, the thickness of the dielectric was controlled to reduce the gate threshold voltage. The high threshold voltage was obtained with a thick dielectric layer of over 1 µm. The thickness of 2:1 PVP-PMF was decreased by decreasing the UA flow rate and keeping the rest of the parameters the same. The thickness optimization comes from the tradeoff between the UA flow rate and continuous line coverage by the ink stream. The UA flow rate was decreased to a minimum of 20 sccm to deposit a pinhole-free dielectric layer. Below this rate, the film was too thin to leak between S/D and gate contacts. The morphology of the dielectric film was good for this UA setting, with uniform and continuous printing.

The devices with thin dielectric layers were characterized, and Figure 9a,b show the transfer and output curves of one transistor, respectively. It can be observed that the device current drops below 1 nA at around a 4 V gate voltage. The on/off ratio of the device was found to be 1.5 × 10^4^, and the device mobility was calculated as 6.1 cm^2^ V^−1^ s^−1^. This performance is a promising result for the low-voltage operation of CNT-based transistors. The low threshold voltage of −4.8 V can be extracted from the linear transfer curve of the device, where the highest slope intersects the gate voltage axis, as shown in Figure 9c.

Figure 10 shows the transfer curve of the same device after 100 days, and it compares it with curves 2 days after printing. The on and off current increased substantially in 100 days, and performance degraded. This could be caused by the disruption of CNT networks and the penetration of ambient species inside the dielectric film. The lifetime of these CNT-/PVP-PMF-based TFTs can be enhanced by encapsulating them with some inert/nonreactive material.

## 4. Discussion

In this work, the printability of cross-linked PVP ink was explored for its use in CNT-based flexible transistors fabricated by AJP. The optimal weight ratio of PVP to PMF in the crPVP solution was found to be 2:1, and the annealing temperature was kept at 170 °C for compatibility with the flexible substrate. Other wt. ratios showed large hysteresis during forward and backward sweeps in transfer curves of fully printed CNT TFTs. For continuous printing and ink stability, it was found that 2.5 wt.% PVP-PMF in PGMEA offers the best-printed film without pinholes. The thickness of the dielectric film was easy to control by changing the UA flow rate when the concentration was 2.5 wt.%. Hence, the thickness of the polymer dielectric was optimized by changing AJP parameters to obtain a thickness below 1 µm, which enables the low-voltage operation of printed devices. It was found that the TFT has a threshold voltage as low as −4.8 V with an on/off ratio of more than 10^4^. Finally, the environmental stability of the device was analyzed over 100 days to show the performance degradation over time. This work investigated the use of polymer dielectrics in fully printed transistors with low hysteresis and a low threshold voltage.

## Figures and Tables

**Figure 1 nanomaterials-12-04487-f001:**
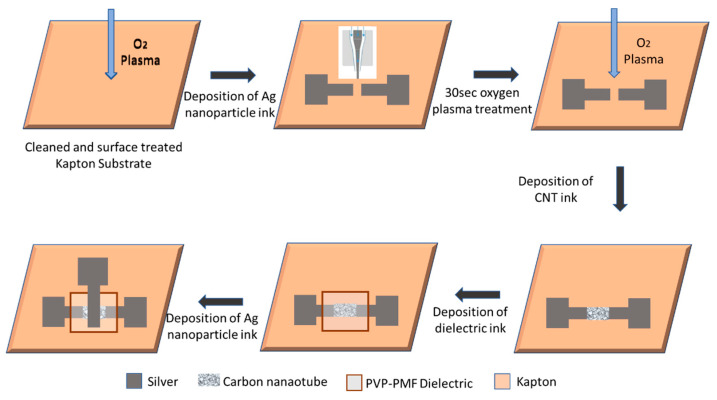
Detailed diagram of CNT TFT fabrication process on Kapton substrate using aerosol jet printing.

**Figure 2 nanomaterials-12-04487-f002:**
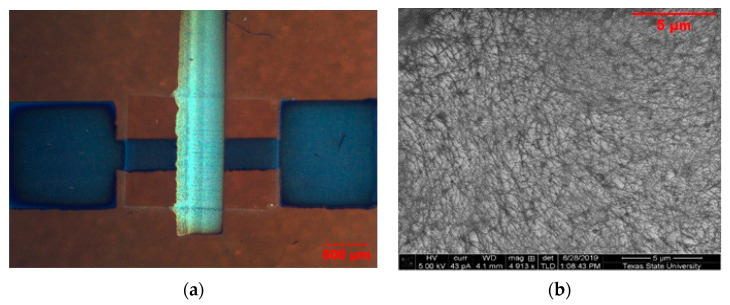
(**a**) Microscope image of printed device on Kapton (**b**); SEM image of the CNT network after 3 layers of printing.

**Figure 3 nanomaterials-12-04487-f003:**
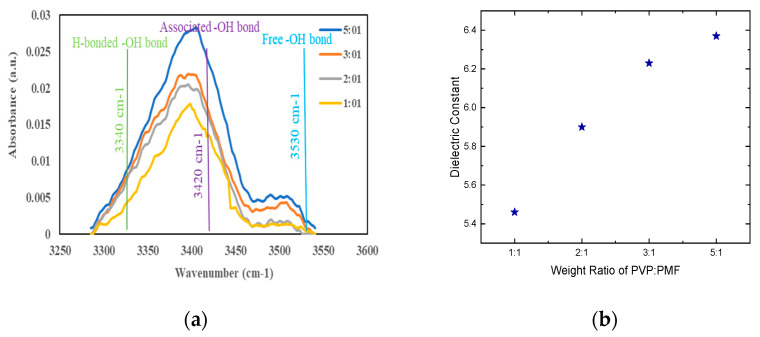
(**a**) FTIR spectra. (**b**) Dielectric constant of PVP-PMF dielectrics at various weight ratios.

**Figure 4 nanomaterials-12-04487-f004:**
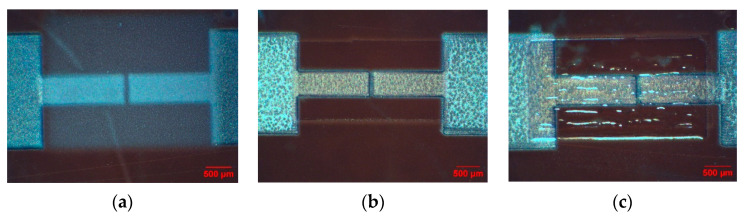
Microscope image of printed CNT source and drain, channel, and dielectric layer on Kapton with various concentrations of 2:1 cross-linked PVP ink and carrier gas pressure: (**a**) 2.5 wt.% solution with UA of 18 sccm; (**b**) 2.5 wt.% solution with UA of 22 sccm; (**c**) 5 wt.% solution with UA of 28 sccm.

**Figure 5 nanomaterials-12-04487-f005:**
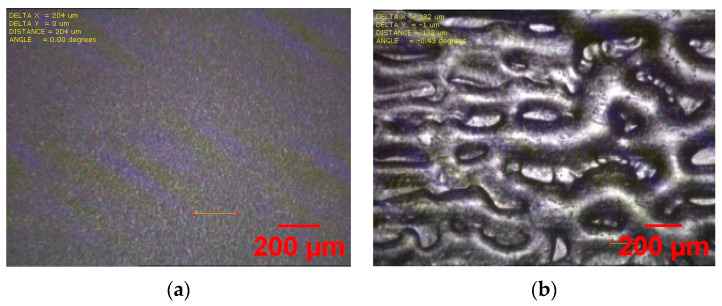
Microscope image of as-printed cross-linked PVP dielectric on Kapton. (**a**) Smooth surface 2:1 wt. ratio 2.5 wt.% solution with UA = 22 sccm; (**b**) surface showing pinhole for 5:1 wt. ratio 4 wt.% solution with UA = 35 sccm.

**Figure 6 nanomaterials-12-04487-f006:**
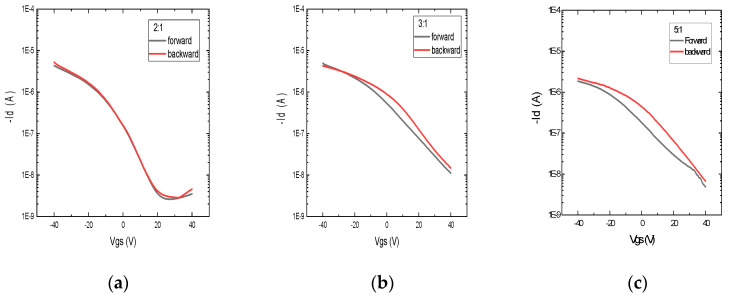
Transfer characteristics showing hysteresis in gate threshold voltage by forward (on → off) and reverse sweeps (off → on) of printed transistors using PVP-PMF dielectrics at various weight ratios: (**a**) 2:1, (**b**) 3:1, and (**c**) 5:1.

**Figure 7 nanomaterials-12-04487-f007:**
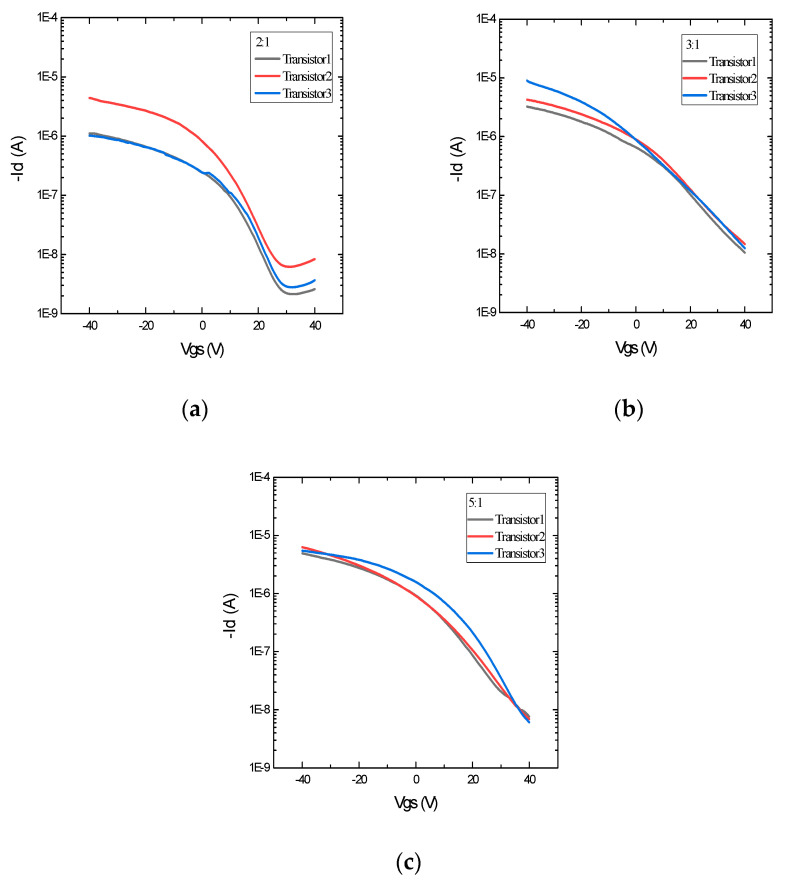
Transfer characteristics of printed devices with PVP-PMF dielectrics at various weight ratios: (**a**) 2:1, (**b**) 3:1, and (**c**) 5:1.

**Figure 8 nanomaterials-12-04487-f008:**
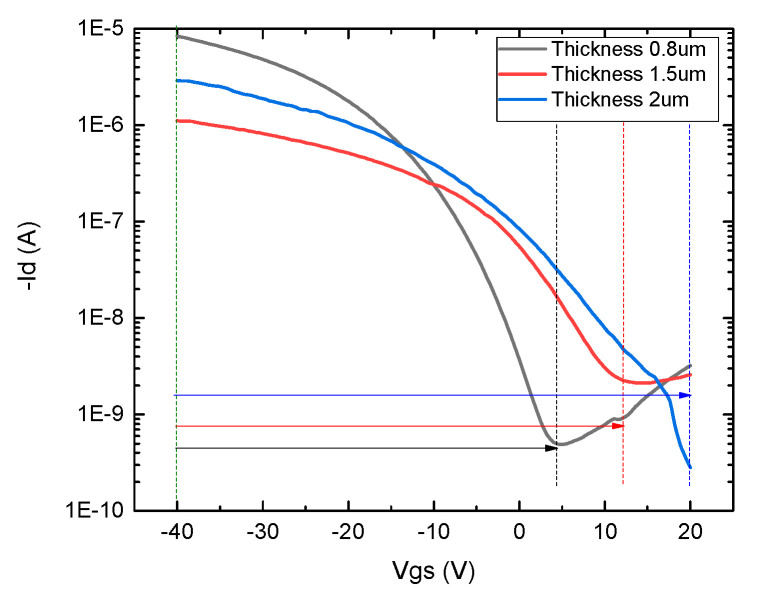
Transfer characteristics of three fully printed CNT TFTs on Kapton with different thicknesses of 2:1 wt.% cross-linked PVP dielectrics (L = 80 µm; W = 500 µm).

**Figure 9 nanomaterials-12-04487-f009:**
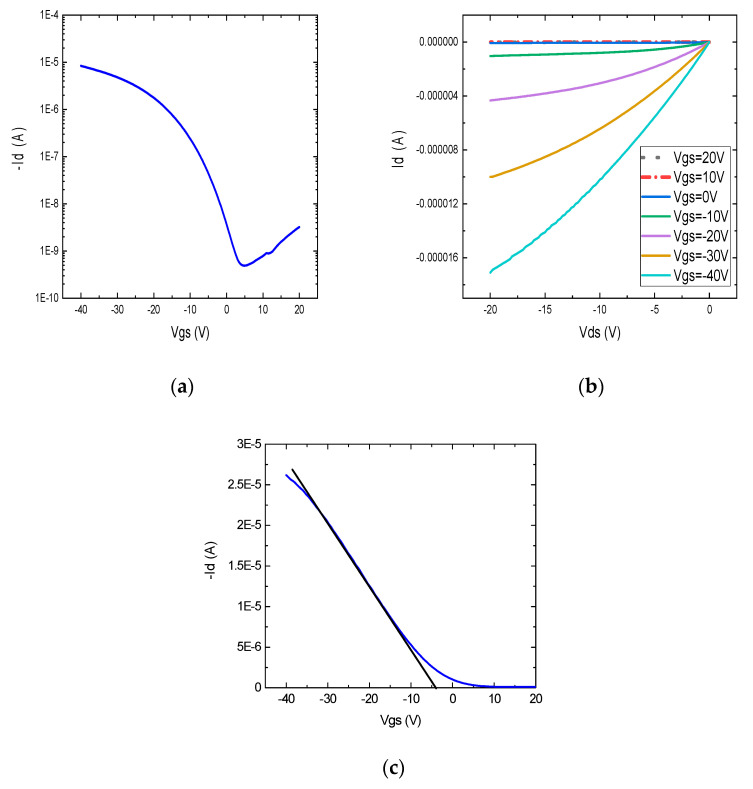
(**a**) Transfer (at Vds = −10 V), (**b**) output characteristics, and (**c**) linear transfer characteristics showing threshold voltage extraction of printed transistor with 2:1 wt. ratio of PVP to PMF (L = 80 µm, W = 500 µm).

**Figure 10 nanomaterials-12-04487-f010:**
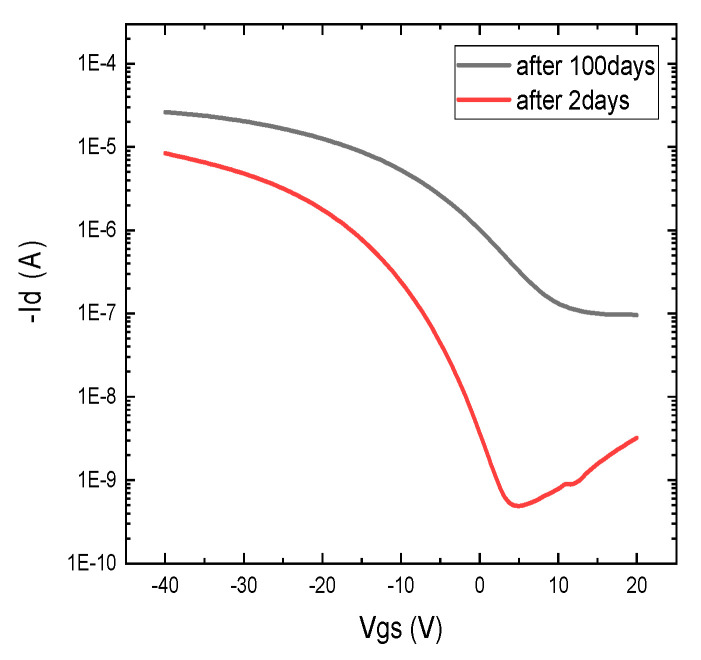
Transfer characteristics of printed device with 2:1 wt. ratio of PVP to PMF after 2 days and 100 days of device printing without encapsulation.

## Data Availability

Not applicable.

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
