# Peer review of "All-Aerosol-Jet-Printed Carbon Nanotube Transistor with Cross-Linked Polymer Dielectrics"

_nanomaterials, 2022, doi:10.3390/nano12244487_

Round 1

Reviewer 1 Report

The authors of the manuscript “Completely printed Carbon nanotube transistor with cross-linked polymer dielectrics for low-voltage operations” described the new one single technique, aerosol jet printing, to fabricate a CNT TFT using cross-linked PVP-PMF dielectric material. CNT networks were deposited utilizing layer-by-layer deposition to decrease bundling and rinsed with solvent to remove excess surfactant. crPVP solutions were prepared at different weight ratios of PVP: PMF and characterized to be used as the dielectric layer. 2:1 wt. ratio of PVP:PMF was found to have less hysteresis in gate threshold voltage for forward and reverse sweeping. This research significantly contribute toward the use of polymer dielectrics for completely printed low voltage operated transistors for low power dissipation. It also add significant data on the best weight ratio of PVP:PMF to reduce hysteresis in printed transistors for reliable operation.

The manuscript is recommended for publication in the journal as presented.

Author Response

Thank you for reviewing the manuscript and giving comments. We really appreciate your valuable feedback.

Reviewer 2 Report

1. The authors insisted on a CNT TFT with crosslinked polymer dielectrics for low voltage operations. However, the corresponding data showing the low voltage operation of the devices were not presented. Thus, such claim or description on the low voltage operation cannot be accepted.

2. In 2017, Cao et al. has already reported on the completely printed, flexible, stable, and hysteresis-free CNT TFTs via aerosol jet printing (https://doi.org/10.1002/aelm.201700057). The authors’ manuscript does NOT effectively deliver a distinctive difference, compared to previous researches.

3. PVP gate insulators with residual -OH groups has been already widely known to cause instability problems in the field of TFTs. Why is the observation of the PVP ratio effect for optimization, using the fully printed CNT TFT platform, critical and challenging?

3. The optimization process was not sufficiently carefully conducted. Does the optimization process need to examine other conditions such as 2.2:1, 1.8:1, and 1.6:1 to find an optimum condition?

4. Investigation of completely printed TFTs is challenging. The present version of manuscript, which emphasizes the widely known PVP ratio effect with the insufficient optimization conditions, is not proper for publication through the journal.

5. Throughout the whole manuscript, there are huge gaps between the authors’ claims and the actually presented data, and the corresponding data or method are not even shown, e.g., the low voltage operation, the dielectric properties improvement, the 4-V threshold voltage extraction, the film morphology, the low power consumption, the 2.5wt% best-printed film without a pinhole, the thickness optimization, and reliable operation.

Author Response

Thank you reviewing the manuscript and giving comments. Please see the attachment.

Round 2

Reviewer 2 Report

The manuscript contains unacceptable claims and erroneous descriptions. The manuscript with the present academic quality cannot be published through the journal. The manuscript must be improved extensively and professionally, from usage of terminology to experimental design and data analysis. Sincerely.

Author Response

Thank you for reviewing the manuscript. Please specify the specific improvements needed broadly.